# Molecular Biomarkers in Peri-Implant Health and Disease: A Cross-Sectional Pilot Study

**DOI:** 10.3390/ijms23179802

**Published:** 2022-08-29

**Authors:** Alejandra Chaparro, Víctor Beltrán, Daniel Betancur, Ye-Han Sam, Haniyeh Moaven, Ali Tarjomani, Nikolaos Donos, Vanessa Sousa

**Affiliations:** 1Department of Periodontology, Laboratory of Periodontal Research, Center for Biomedical and Innovation Research, Faculty of Dentistry, Universidad de Los Andes, Santiago 7620001, Chile; 2Center of Investigation and Innovation in Clinical Dentistry, Faculty of Dentistry, Universidad de La Frontera, Temuco 4780000, Chile; 3Periodontology Unit, Department of Surgical Stomatology, Faculty of Dentistry, Universidad de Concepción, Concepción 4070386, Chile; 4Centre for Host-Microbiome Interactions, Periodontology and Periodontal Medicine, Faculty of Dentistry, Oral and Craniofacial Sciences, King’s College London, Guy’s Hospital, London SE1 9RT, UK; 5Centre for Oral Clinical Research, and Centre for Oral Immunobiology & Regenerative Medicine, Institute of Dentistry, Barts and The London School of Medicine and Dentistry, Queen Mary University London, London E1 4NS, UK

**Keywords:** peri-implantitis, CCL-20/MIP-3α, BAFF/BlyS, IL-23, RANKL, Osteoprotegerin

## Abstract

Background: The aim of this feasibility study was to investigate the concentration level of CCL-20/MIP-3α, BAFF/BLyS, IL-23, RANKL, and Osteoprotegerin in the Peri-Implant Crevicular Fluid (PICF), from patients diagnosed with peri-implant mucositis and peri-implantitis, and to compare them with PICF from patients with healthy implants. Methods: Participants with at least one dental implant with healthy peri-implant tissues, peri-implant mucositis, or peri-implantitis were included. PICF was collected using paper strips from healthy and diseased peri-implant sites (*n* = 19). Biomarker levels were analyzed using a custom Multiplex ELISA Assay Kit. Results: In comparison to peri-implant health, the peri-implant mucositis group showed an increased concentration of CCL-20 MIP-3α, BAFF/BLyS, IL-23, RANKL, and Osteoprotegerin. The peri-implantitis group had the lowest median concentration of Osteoprotegerin (1963 ng/mL); this group had a similar concentration of RANKL (640.84 ng/mL) when compared to the peri-implant health group. BAFF/BLyS (17.06 ng/mL) showed the highest concentration in the peri-implantitis group. Conclusions: This feasibility study suggests that IL-23 and RANKL may help to elucidate the pathogenesis during the conversion from peri-implant health to peri-implantitis. Further research is required in BAFF/BLyS for the early diagnosis of peri-implantitis.

## 1. Introduction

Peri-implantitis has been defined as a plaque-associated pathological condition around dental implant tissues, it is characterized by inflammation of the peri-implant mucosa and progressive loss of supporting bone [1]. Prevalence of peri-implantitis has been reported in up to 47% of patients [2].

Several studies have associated previous presence of periodontitis with increased prevalence of peri-implantitis [3] with OR = 3.6 (95% CI: 1.7–7.6) [4], and similar to periodontitis, the development of a pathogenic biofilm and the initiation of peri-implant disease has been established [5,6]. Evidence has shown that the microbiota associated with peri-implantitis is complex [7], comprising bacteria, virus, fungi, and archaea, which may lead to a higher inflammatory response and osteoclastic activity [8]. Thereafter, the disturbance in the host-microbiome relationship, initiated by the release of bacterial products and inflammatory mediators in circulation system, will lead to activation of an acute followed by a chronic inflammatory response by the host [9].

As the sensitivity of clinical and radiographic diagnosis of peri-implant inflammatory disease is low (detection only once the tissue damage has already taken place), current research is directed towards supplementary, non-invasive techniques by which early detection of pathological biological activities can be detected. Evaluation of the level of inflammatory biomarkers may help to accurately detect early disease onset and to monitor disease activity [10]. The Peri-Implant Crevicular Fluid (PICF) in peri-implant crevice contains valuable diagnostic biomarkers, microorganisms, and host cells, that may reflect the physiological interactions between the host and the microflora. Biomarkers can then be quantified in a reproducible manner. Different techniques have been utilized to collect and analyze PICF samples for diagnosis purposes in peri-implant disease assessment [10]. IL-23 is an inflammatory cytokine (CK) that regulates T-helper 17 cells (Th-17) maintenance and expansion. This CK is mainly secreted by activated monocytes, macrophages, and dendritic cells. IL-23 will promote Th-17 cells to produce IL-17 which subsequently will stimulate Receptor Activator of Nuclear factor Kappa-Β Ligand (RANKL) [11].

RANKL (ligand) is expressed by osteoblasts and bone marrow stromal cells and adheres to the RANK receptor on mature osteoclast and osteoclast precursor cells to promote its differentiation [12]. Besides osteoclastic activity induction, RANKL induce osteoclast attachment to bone surface [13] and their longevity [14].

RANKL may also bind to Osteoprotegerin, a protein which is secreted by osteoblasts and is a potent inhibitor of osteoclast formation. The mechanism is by preventing binding of RANKL to RANK [11] in which is shown to be associated with the process of bone resorption [15,16]. Chemokine ligand-20 (CCL-20) is strong chemotactic agent for B and T lymphocytes and weakly attracts neutrophils [17]. B-cell Activating Factor (BAFF, also known as BLyS) is a member of the Tumor Necrosis Factor (TNF) ligand family that plays an important role in B-lymphocyte differentiation, maturation and survival. B-cell chemotaxis and proliferation in periodontal tissues leads to an increased chronic inflammatory response in periodontal tissue facilitates alveolar bone destruction as these cells express RANKL in periodontal tissue [17].

The quantification of the humoral factors’ concentrations in the PICF may be beneficial for the assessment of stages from peri-implant health or disease, and for early detection of peri-implantitis. Therefore, the aim of this feasibility study was to investigate the concentration level of CCL-20/MIP-3α, BAFF/BLyS, IL-23, RANKL, and Osteoprotegerin in the Peri-Implant Crevicular Fluid (PICF), from patients diagnosed with peri-implant mucositis and peri-implantitis, and to compare them with PICF from patients with healthy implants.

## 2. Results

### 2.1. Demographic Data

All participants were medically fit and well. Demographic data of the participants in terms of gender, age, smoking status, and implant position, are summarized in Table 1. The mean age range of the participants in the three different groups was similar: 73.8 years old for peri-implant health, 61.5 years old for peri-implant mucositis, and 67.8 years old for peri-implantitis group. In terms of smoking status; 42% of participants diagnosed with peri-implant health were smokers, all participants with peri-implant mucositis were smokers, whilst 30% of participants diagnosed with peri-implantitis were smokers.

### 2.2. Characteristics of the Sample

Clinical characteristics of the PICF sample collection site are shown in Table 2. In peri-implant health, 57% of the implants were in premolar area and the rest were in mandibular and maxillary first and second molar area. In peri-implant mucositis group, all the implants were in maxillary molar area. In peri-implantitis group, only one implant was from maxillary anterior segment, 40% from maxillary and mandibular premolar, and 50% from posterior segment. All sites with diagnosis of peri-implant mucositis and peri-implantitis had positive BOP. Only 10% of implants with peri-implantitis had positive suppuration on gentle probing. None of the implants included in the study were mobile.

### 2.3. Biomarker Concentration per Group

Descriptive results of biomarker concentration (ng/mL) for different clinical scenarios have been summarized in Table 3. Mean concentration of CCL-20/MIP-3α in healthy sites was the lowest (33.24 ng/mL) when compared to peri-implant mucositis and peri-implantitis. Its concentration increased during initial inflammatory phase (peri-implant mucositis 82.09 ng/mL) in PICF and then reduced to about half in the peri-implantitis stage (42.79 ng/mL) of the disease.

Overall, BAFF/BLyS concentration had an increasing pattern from healthy (9.7 ng/mL) to peri-implantitis (17.065 ng/mL) sites.

The concentration of IL-23 was higher (more than double) in peri-implant mucositis (193.42 ng/mL) compared to healthy (70.16 ng/mL) sites. In peri-implantitis (134.905 ng/mL) sites, there was less concentration compared to peri-implant mucositis, but still higher than peri-implant healthy sites.

RANKL showed a steady high concentration across conditions. Its concentration was recorded highest in peri-implant mucositis (830.35 ng/mL) group followed by peri-implantitis (640.84 ng/mL) and lastly least concentration in peri-implant healthy (629.5 ng/mL) sites.

Osteoprotegerin exhibited the highest recorded concentration in all 3 clinical conditions. Its concentration was recorded highest in peri-implant mucositis group with 2481.5 ng/mL and the lowest in peri-implantitis (1963 ng/mL) group.

## 3. Discussion

Peri-implant disease results from the inflammatory response to pathogenic biofilms which results in soft tissue inflammation and leads to progressive loss of supporting bone. As conventional clinical diagnostic methods (PPD, BOP, and radiographic assessments) have a low sensitivity (ability of a test to correctly identify sites with disease, true positive) and specificity (ability of the test to correctly identify sites without disease, true negative), and also have low predictability value for estimation risk of future progression of disease. Thus, novel, non-invasive techniques are required to early detect presence of inflammation and/or monitor the effect of therapeutic measures [18,19]. Although biological markers can be measured accurately and are reproducible indicators to monitor disease signs on molecular level, more studies are required in this field to assess effectiveness of biomarkers in terms of prediction/monitoring of disease [20].

In this feasibility study, CCL-20/MIP-3α, BAFF/BLyS, IL-23, RANKL, and Osteoprotegerin concentrations were assessed in patients diagnosed with peri-implant health, peri-implant mucositis and peri-implantitis. The outcomes showed that higher concentration of CCL-20/MIP-3α, BAFF, IL-23, and RANKL was present in sites with peri-implant mucositis/peri-implantitis compared to healthy peri-implant tissues.

RANKL is one of the most extensively studied biomarkers in relation to bone metabolism and has been shown to be a valuable biomarker for determination of bone destruction in periodontal and peri-implant disease [10,21]. The results of the current study regarding RANKL level in PICF is in line with available evidence which showed increased expression of RANKL in sites with peri-implant mucositis and peri-implantitis compared to healthy sites [22,23]. These results indicate that RANKL could be one of the biomarkers which can be used to monitor potential progression of peri-implant diseases.

Regarding Osteoprotegerin (OPG) concentration in PICF in different states of health and disease, it was shown that its concentration in established disease (peri-implantitis) is lower (1963 ng/mL) than healthy sites (2069 ng/mL) and also compared to peri-implant mucositis (2481.5 ng/mL). The results of this study are in agreement with previous studies evaluating the concentration of OPG [21,24]. OPG’s value and capacity as diagnostic biomarker should be further explored in additional studies with larger sample sizes to establish the link for this biomarker and possibly to determine the biomarker activity map.

CCL-20/MIP-3α is one of newly targeted biomarkers investigated about its association with peri implant inflammatory disease as its relationship with Th17 cell migration to the site and regulation of humoral immunity [25]. Moreover, it has been showed that some periodontal pathogens (i.e., *Aggregatibacter actinomycetemcomitans*) could increase the expression of CCL-20 in different immune cell types [26]. The over expression of CCL-20 is accompanied by upregulation of CCR6 (its ligand) on neutrophils leading to an increased chemotactic response of these cells [14], which could impact the periodontal and peri-implant inflammatory status [26]. The results of this study failed to show significant difference in concentration of this biomarker in states of health and disease and more controlled studies with larger sample size are required to further investigate and confirm biologic plausibility of CCL-20 in peri-implant inflammatory disease.

BAFF/BLyS is a biomarker that has been investigated in periodontitis patients with systemic inflammatory conditions such as Rheumatoid Arthritis (RA) [27]. Studies have shown increased serum level of BAFF/BLyS in RA patients (compared to healthy patients) despite long term intake of anti-inflammatory medications [27]. It seems that the TNF-α biomarker family may contribute to greater disease activity [21,28]. The results of this study are in agreement with previous clinical and experimental studies showing increased expression of lymphocyte B stimulatory cytokines (BAFF/BLyS) in sites with inflammatory periodontal and peri-implant bone destruction [21,29]. Moreover, there is evidence that BAFF/BLyS play critical roles in naive B cell survival, increasing the expression of RANKL, which could lead to increased osteoclastic activity and subsequent bone resorption [21,29].

IL-23 has been shown to play role in regulation of T-helper 17 cells in animal and human studies [30]. Th-17 cells are further involved in pro-inflammatory response regulation leading to recruitment of neutrophils and macrophages. The evidence suggests that there is an increased expression of Th-17 and IL-23 in sites with peri-implantitis [31]. These findings are in agreement with the results of our study, as we have showed an increased expression of IL-23 in peri-implant mucositis and peri-implantitis compared to healthy sites.

This feasibility clinical study presents with a number of limitations which reduces the impact of the findings and the assumption of general applicability in larger populations. More specifically, the feasibility nature of its design linked with the small sample size, creates difficulties in generalization of the molecular and clinical findings. As such, the results of this study should be interpreted with caution.

Nevertheless, the strength of this feasibility study lies in the results showing that IL-23 and RANKL may help to elucidate the pathogenesis during the conversion from peri-implant health to peri-implantitis; and the potential use of BAFF/BLyS for the early diagnosis of peri-implantitis. In addition, the levels of these biomarkers are probably influenced by multifactorial conditions, including host susceptibility, plaque control, smoke, different kind of abutment connections (and potential different bacterial leakage) and the prosthetic materials of the rehabilitation [32,33,34]. Furthermore, differences in implant design and in the definitive rehabilitation may affect the potential risk of colonization of oral pathogens in the fixture-abutment interface, which could attract to inflammophilic bacteria and activate neutrophils and others immune cells to release chemokines and inflammatory mediators involved in the tissular destruction observed in peri-implantitis. Nevertheless, the strength of the present study lies in proposing IL-23, RANKL, BAFF/BLyS axis, and OPG as potential biomarkers of peri-implant disease. However, our preliminary results need validation with further prospective studies.

## 4. Material and Methods

This feasibility study was designed following CONSORT [35] and STROBE guidelines [36]. Participants were recruited from the Implant Clinic of the School of Dentistry at the Universidad de La Frontera, Temuco, Chile. The study comprised a total of 19 PICF samples derived from sites diagnosed with: peri-implant health (7), peri-implant mucositis (2), and peri-implantitis (10). Inclusion criteria were participants over 18 years old; presence of at least one dental implant accessible to peri-implant probing; and participants presenting with healthy peri-implant tissues, or peri-implant mucositis, or peri-implantitis. Case definition for clinical scenarios (peri-implant health, peri-implant mucositis, and peri-implantitis) was based on the 2017 EFP/AAP classification system for peri-implant health and diseases [37]. Exclusion criteria included participants suffering from chronic inflammatory diseases (diabetes, cardiovascular, chronic inflammatory, autoimmune, and infectious diseases), and systemic or topical anti- microbial/anti-inflammatory therapy for the previous three months. Sample collection was performed following a standardized protocol [20]. Participants were asked to avoid eating, drinking, and oral hygiene measurement 30 min prior to sample collection. Briefly, the supra-gingival plaque biofilm was carefully removed using curettes without contacting the marginal gingiva. The collection sites were isolated using cotton rolls and dried gently with air jets. PICF samples were subsequently obtained using sterile PerioPaper™ strips (Oraflow Inc., New York, NY, USA). Strips were inserted 3–5 mm into the sulcus/pockets for 30 s. Strips contaminated by saliva and/or blood were discarded. Samples were then kept in sterile 1.5 mL tubes at −80 °C until elution. For elution of PICF, 4 strips per implant were placed into a 1.5 mL tube containing 160 μL of phosphate buffer saline (PBS) (Corning, Media tech Inc., New York, NY, USA) and protease inhibitor cocktail (EDTA Complete TM, mini, EDTA-free Protease Inhibitor Cocktail, Roche, Mannheim, Germany). Tubes were vortexed and incubated on ice for 30 min, and then centrifuged at 12,000× *g* for 5 min at 4 °C. The eluate was collected and placed on ice. The elution procedure was repeated and both eluates were pooled and stored at −80 °C until needed for analysis.

CCL-20/MIP-3α, BAFF/BLyS, IL-23, RANKL, and Osteoprotegerin concentrations in PICF samples were quantified using a designed custom kit for Multiplex Enzyme-Linked Immunosorbent Assays (ELISA) according to manufacturer instructions. All samples were analyzed in duplicate. The re-suspended microsphere cocktail (50 μL) was added to each well of a 96-well black plate. PICF eluate (50 μL) was added to each well. The final concentration of the samples was calculated using a Milliplex Analyst (version 5.1; Merck KGaA, Darmstadt, Germany).

The statistical analysis for this study was performed using STATA software (version 15.1; StataCorp, College Station, TX, USA). Descriptive statistics were performed for categorical variables, which were described through frequencies and percentages. Continuous variables were described through the median and interquartile range.

## Figures and Tables

**Table 1 ijms-23-09802-t001:** Demographic data of participants in the study including number of samples per group, tooth position, age range, and smoking status Each participant may present different conditions of health or peri-implant disease from the total of their implants present in mouth and in function.

Variable		Peri-Implant Condition		
		Peri-Implant Health (*n* = 7)	Peri-Implant Mucositis (*n* = 2)	Peri-Implantitis (*n* = 10)
Gender	Female	4	0	8
	Male	3	2	2
Age	Range	67–78	56–67	48–78
	Mean	73.8	61.5	67.8
	Median	75	61.5	67.5
Smoking Status	Non-smoker	4	-	7
	Former Smoker	-	-	-
	Current smoker	3	2	3
Implant position (FDA tooth numbering system)		45, 16, 14, 25, 37, 16, 14	17, 16	45, 46, 47, 47, 46, 36, 35, 25, 24, 21

**Table 2 ijms-23-09802-t002:** Summary of clinical characteristics of the PICF sample collection site. Variables: Bleeding on Probing (BOP), Suppuration, Probing Pocket Depth (PPD), Clinical Attachment Level (CAL), and Mobility.

Variable		Peri-Implant Condition		
		Peri-Implant Health (*n* = 7)	Peri-Implant Mucositis (*n* = 2)	Peri-Implantitis (*n* = 10)
		Frequency (%)	Frequency (%)	Frequency (%)
Location in arch	Incisor/canine	0 (0%)	0 (0%)	1 (10%)
	Premolar	4 (57%)	0 (0%)	4 (40%)
	Molar	3 (43)%	2 (100%)	5 (50%)
BOP (+)		0 (0%)	2 (100%)	10 (100%)
Suppuration (+)		0 (0%)	0 (0%)	1 (10%)
PPD (Median)		3.39 mm	4.15 mm	4.25 mm
CAL (Median)		4.29 mm	5.36 mm	5.53 mm
Mobility (+/−)		0	0	0

**Table 3 ijms-23-09802-t003:** Biomarkers concentration (median and interquartile range) from PICF across different peri-implant conditions.

Biomarker Concentration Range Median (ng/mL)	Peri-Implant Condition		
	Peri-Implant Health (*n* = 7)	Peri-Implant Mucositis (*n* = 2)	Peri-Implantitis (*n* = 10)
CCL-20/MIP-3α	11.45–103.84 (33.24)	29.45–134.73 (82.09)	13.25–149.26 (42.79)
BAFF/BLyS	3.48–27.06 (9.7)	10.36–17.23 (13.79)	8.69–47.45 (17.065)
IL-23	20.49–405.97 (70.16)	86.79–300.05 (193.42)	55.7–234.57 (134.905)
RANKL	595.42–1168 (629.5)	612.46–1048.24 (830.35)	455.48–1003.12 (640.84)
Osteoprotegerin	1474.88–6071 (2069)	1970–2993 (2481.5)	885.88–4086 (1963)

## Data Availability

Not applicable.

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
