# Peer review of "Molecular Biomarkers in Peri-Implant Health and Disease: A Cross-Sectional Pilot Study"

_ijms, 2022, doi:10.3390/ijms23179802_

Round 1

Reviewer 1 Report

Pros: This is a timely feasibility study exploring molecular biomarkers which may be useful for the early diagnosis of peri-implantitis. Given that almost half of implant patients may develop peri-implantitis, it is important that this knowledge base be expanded and thus this study is welcomed.

Cons: Small sample sizes for all groups, particularly for peri-implant mucositis; although this reviewer is unsure of whether this may be due to the actual availability of patients with implants (considering their patient pool size?), as that was not discussed in the manuscript.   

Suggested changes and corrections:

For consistency and accuracy, please add a "/" between CCL-20 and MIP-3alpha throughout (as you have used for BAFF/BLyS).

line 46- "fungi" not fungus.

line 68- Osteoprotegerin (OPG) is secreted by osteoblasts, (not just by osteoblast cell lines).

Table 1- it is not clear whether all/most of the subjects had only 1 implant, or if only one of their implants was healthy (or had peri-implant mucositis/peri-implantitis); please clarify.

Table 2- For consistency, use abbreviations for all; i.e., add BOP -and also define PPD and CAL.

line 156- 2069 is wrong, correct to 2481 as in Table 3

line 179- BAFF/BLyS leads to improved survival of B-cells isolated from in vivo; not just in "cell line" so it is not correct to only use "cell line"

line 197- A last sentence should be added stating something like "This possibility needs to be supported with further studies."

Author Response

Dear reviewer. Thanks for the suggestions, they were all addressed and changes were incorporated to the manuscript accordingly. 

Reviewer 2 Report

In my opinion:

- lack of graphic presentation of research results,

- too few patients in particular groups.

Author Response

Dear reviewer, thanks for your comments and your review. At the moment of writing considered the way the information was presented is sufficient to deliver the data in a clear manner to the reader. For further work in relation to the results obtained in this study we will certainly add more sophisticated graphic support and, since we will have a longer execution time for the research, more participants can be enrolled. The limitations from the current study have been added to the discussion between lines 198 and 202.

Reviewer 3 Report

Dear Authors 

the paper is well written and can be considered for publication after minor revisions

In particular some topics are need to be discussed before acceptance

1) Please discuss the evaluation of such markers in implants with different connections (and potential different bacterial leakage). Please cite PubMed ID26922985

2) At the same way, discuss the possibility to extend your findings in implants supporting rehabilitation with different prosthetic materials like zirconia. Please cite PubMed ID 344256653) Moreover, can your markers hypothesize a different oral hygiene follow-up according to theier values? Please cite PubMed ID28696070

Author Response

Answer to suggestion 1) and 2): Dear reviewer, this study represents the first approach to explore the levels of the studied biomarkers in the peri-implant crevicular fluid (PICF) and their relationship with the peri-implant diagnosis and other conditions related with the implant. We are very enthusiastic about the preliminary results and will continue testing these biomarkers with different connections and prosthetic materials in the future. In addition, as you kindly suggested, we included the recommended references in the discussion.